# Evaluation of Near-Infrared Reflectance and Transflectance Sensing System for Predicting Manure Nutrients

Xiaoyu Feng *, Rebecca A. Larson and Matthew F. Digman

Department of Biological Systems Engineering, University of Wisconsin-Madison, 201 Agricultural Engineering Building, 460 Henry Mall, Madison, WI 53706, USA; rebecca.larson@wisc.edu (R.A.L.); digman@wisc.edu (M.F.D.)
* Correspondence: xfeng43@wisc.edu; Tel.: +1-864-986-2404

**Abstract:** Livestock manure is widely applied onto agriculture soil to fertilize crops and increase soil fertility. However, it is difficult to provide real-time manure nutrient data based on traditional lab analyses during application. Manure sensing using near-infrared (NIR) spectroscopy is an innovative, rapid, and cost-effective technique for inline analysis of animal manure. This study investigated a NIR sensing system with reflectance and transflectance modes to predict N speciation in dairy cow manure using a spiking method. In this study, 20 dairy cow manure samples were collected and spiked to achieve four levels of ammoniacal nitrogen ($NH_4$-N) and organic nitrogen (Org-N) concentrations that resulted in 100 samples in each spiking group. All samples were scanned and analyzed using a NIR system with reflectance and transflectance sensor configurations. NIR calibration models were developed using partial least square regression analysis for $NH_4$-N, Org-N, total solid (TS), ash, and particle size (PS). Coefficient of determination ($R^2$) and root mean square error (RMSE) were selected to evaluate the models. A transflectance probe with a 1 mm path length had the best performance for analyzing manure constituents among three path lengths. Reflectance mode improved the calibration accuracy for $NH_4$-N and Org-N, whereas transflectance mode improved the model predictability for TS, ash, and PS. Reflectance provided good prediction for $NH_4$-N ($R^2 = 0.83$; RMSE = 0.65 mg mL$^{-1}$) and approximate predictions for Org-N ($R^2 = 0.66$; RMSE = 1.18 mg mL$^{-1}$). Transflectance was excellent for TS predictions ($R^2 = 0.97$), and provided good quantitative predictions for ash and approximate predictions for PS. The correlations between the accuracy of $NH_4$-N and Org-N calibration models and other manure parameters were not observed indicating the predictions of N contents were not affected by TS, ash, and PS.

**Keywords:** NIR; manure sensing; nitrogen; spiking; dairy cow

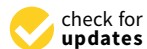



## 1. Introduction

Nutrients contained in animal manure are a valuable resource for crop growth when applied at agronomic rates. However, overapplication can lead to increased nutrient losses to the environment. Variability in manure nutrient concentrations [1] and traditional chemical analysis of manure composition [2] makes precise nutrient application difficult to achieve. Currently, livestock farmers use agitation and sampling techniques to manage the variability in manure nutrients [1,3]. However, real-time nutrient sensing has the potential to improve the accuracy and precision of applied nutrients and enhance crop productivity.

Near-infrared (NIR) spectroscopy is a high-energy vibrational spectroscopy performed in the wavelength range between 750 to 2500 nm where a linear relationship between absorbance and concentration is exhibited by most biological and agricultural materials [4]. It provides an alternative analytical technique to traditional analysis methods for determining the composition of animal manure based on the relationships between spectral and compositional properties of a set of samples. Traditional laboratory manure chemical analysis methods can be expensive (particularly for large sample numbers) and take significant time to obtain results. This time lag results in concentration data being available after

manure application, allowing for changes in nutrient accounting systems but not the actual application rate. A NIR manure sensing system has advantages over traditional analyses as it is rapid, has high efficiency, is non-destructive of samples, and avoids the use of hazardous chemicals [5].

Most of the existing literature investigates spectral data from NIR systems to analyze nutrient concentrations in soil [6,7] and agricultural waste [8–15] with promising results using a reflectance mode. For example, Raju et al. [15] used a NIR spectrometer with a diffuse reflectance probe to monitor the total ammoniacal nitrogen ($NH_4$-N) of an anaerobic cattle manure digester and the NIR model provided excellent predictions of $NH_4$-N for monitoring and screening purposes. Ye et al. [13] collected various types of manure samples and obtained spectra data using a NIR spectrometer. The results presented good predictions for total nitrogen (TN) of animal manure. Malley, Yesmin, and Eilers [14] used a Foss NIR system in the reflectance mode to predict constituents in hog manure and hog manure-amended soil and concluded the NIR spectroscopy in reflectance mode was a feasible technology for analyzing moisture, organic matter, nitrogen (N), and phosphorus (P) in hog manure. Kemsley et al. [10] compared three spectroscopic techniques for the determination of $NH_4$-N in composted manure samples and found the NIR reflectance had the best performance. Currently, only a few researchers have investigated the feasibility of NIR spectroscopy for manure composition analysis in transflectance mode. Saeys et al. [16] compared the reflectance and transflectance presentation modes for nutrient content analysis in hog manure and the results indicated the transflectance mode performed better than the reflectance mode for predicting TN, $NH_4$-N, dry matter (DM), and other constituents measured in the study.

Although the NIR spectroscopy has been demonstrated as a reliable technique in manure nutrient analysis, the sensing systems require periodic maintenance due to the large variations in manure which are expensive, time-consuming, and technically difficult [17]. Additionally, as the NIR analysis of manure compositions is developed based on the relationship between nutrient concentrations and the spectral data, it is unknown whether the changes in the NIR spectral data are driven by the direct variance of N concentration or by the indirect variance of other manure constituents. This study explores the changes of NIR predictions by adding external N sources into raw manure samples without affecting other manure characteristics. Several spiking strategies have been suggested to analyze soil properties using NIR spectroscopy for establishing calibration models [18–23]. Yet, no research has been reported to include the spiking methodology in developing calibration models of NIR in manure analysis.

We hypothesized that (1) the transflectance mode of NIR has a better performance than the reflectance mode for predicting manure constituents; (2) NIR spectrum is directly influenced by N concentrations in manure samples and independent to other manure characteristics. In order to examine the hypothesis, calibration models of NIR sensing systems for prediction N speciation and other manure constituents were constructed using dairy cow manure in this study. Calibration samples were prepared by collecting manure on dairy farms and spiking the raw samples with adding external N sources. Additionally, these samples were used to determine whether the changes in the NIR spectral data were driven by the direct variance of N concentration or by the indirect variance of other manure constituents. Finally, the study also evaluated sensor configurations of a reflectance contact sensor and a transflectance probe with three different optical path lengths (1, 2, and 5 mm) to assess the configuration that is most effective in predicting manure constituents.

## 2. Material and Methods

### 2.1. Manure Sampling and Chemical Analysis

A set of 20 manure samples were collected and used for model calibration, labeled from S1 to S20. Dairy cow manure samples were collected from the six dairy farms in Wisconsin between October 2020 and May 2021. Approximately 2 L of each sample was collected then stored in closed containers at 4 °C for less than 7 days until subsampling

and analyses were performed. During subsampling, manure was mixed on a stirring plate for 1 h and a 500 mL subsample was prepared for certified laboratory analysis, a 30 mL subsample was separated for particle size analysis, and ten 50 mL subsamples were prepared for N spiking assessments.

Particle size distribution of manure samples was analyzed following a modified procedure of ASTM D6913 [24]. Bulk samples (30 mL) were placed in ceramic dishes and dried in oven at 105 °C for 24 h. Particle size fractioning of the dried manure sample was performed using a RETSCH AS 200 control vibratory sieve shaker (Retsch GmbH, Garman) with six stacked mesh sieves (53, 125, 250, 500, 2000, and 4000 μm). The total weight of each dried manure sample was measured before separation and the dry matter collected on each sieve after separation.

Chemical analyses of the manure samples for total solids (TS), TN, $NH_4$-N, P, K, and ash was performed at University of Wisconsin Soil and Forage Laboratory in Marshfield, WI, a certified lab by the Minnesota Department of Agriculture in the Manure Analysis Proficiency (MAP) Program, following the A3769 standard [25].

### 2.2. Manure Spiking

Subsamples of manure were used for N spiking. One subsample was used as a reference that did not receive any additional N compounds. The remaining subsamples received incremental $NH_4$-N or Org-N additions by dissolving a specific amount of ammonium chloride ($NH_4Cl$) or Arginine in the sample. Each raw sample was spiked at four levels which were 1.25, 1.5, 2, and 4 times the N concentration analyzed by the certified lab of the raw manure. The concentration of Org-N in manure sample was calculated by subtracting the $NH_4$-N from TN (Org-N = TN − $NH_4$-N). The $NH_4$-N and Org-N for spiking were prepared using ammonium chloride ($NH_4Cl$) and Arginine, respectively. All spiking samples were shaken by inverting 20 times and stored in a refrigerator at 4 °C for 24 h until the chemical dissolved completely before scanning with the NIR sensors for analysis.

### 2.3. Near-Infrared (NIR) Spectroscopic Device and Measurement

A total of 200 samples, including both $NH_4$-N and Org-N spiking groups, were scanned in the laboratory using a near-infrared spectrometer (Model PSS-2120, Polytec, Waldbronn, Germany). Each sample was scanned by one reflectance contact head sensor (HS) (Model PSS-H-B01, Polytec, Waldbronn, Germany) and one transflectance liquid probe (Model Falcata XP 6, Hellma Analytics, Müllheim, Germany) assembled with three different optical path lengths of 1, 2, and 5 mm (PB1, PB2, PB5), respectively. The reflectance (R) measured by HS and transmittance (T) measured by PB was converted to relative absorbance values using the log (1/R) and log (1/T) transformations, respectively. The reference spectrum of head sensor was pure white, while the transflectance probes were referred to distilled deionized (DDI) water to obtain better results. Spectra were recorded over a wavelength range of 1100 to 2100 nm with a step of 2 nm under reflectance and transflectance modes. A specific Polytec NIR Sample Cup with optical quartz bottom that was nearly completely transparent was used in reflectance mode. Manure was manually stirred before pouring into the cup and between successive measurements to avoid solid settling. However, there was no stirring during the measurement to avoid the scanning noise of contact sensor surface. In transflectance mode, manure was contained in a plastic sample container and stirred before and during the measurement. Three measurements were taken for each sample using each sensor and the average of measurements was calculated for further analysis.

### 2.4. Statistical Analysis and PLS Calibration Assessment

Descriptive statistics and regression analysis were performed using RStudio Version 1.2.1335. The acquisition of spectroscopic measurement data was achieved using the Polytec analytic software (PAS LABS Version 1.2). All calculations and analysis were performed with MATLAB version 9.8 R2020a (MathWorks, Inc., Natick, MA, USA), PLS_Toolbox

version. 8.9 with MATLAB [26], and Microsoft Excel version 16.0 (Microsoft Corporation, Redmond, WA, USA). NIR calibrations models were developed using partial least-square (PLS) regression to establish the relationships between spectral data and chemical reference values of each parameter. Successful predictions of nutrient content using PLS method depends on proper calibration methods including preprocessing methods and multivariate methods for regression analysis and the appropriate variable selection techniques [6]. Therefore, pre-processing techniques including no pre-treatment, mean center (MC), multiplicative scatter correction (MSC), Savitzky-Golay smoothing with first and second derivation (D), and combinations of any two of them were applied to process the spectral data. Cross-validation (CV) was used based on the sample size of 100 for each spiking group [27]. The k-fold method of venetian blinds was selected for CV with 10 data splits resulting in 10 samples per sub-validation dataset. The optimum number of latent variables (LV) of PLS regression was determined by the software automatically.

Statistical indicators of coefficient of determination ($R^2$), root mean square error (RMSE), and the ratio of performance to deviation (RPD) of NIR calibration results were considered to assess the model performance in this study. $R^2$ indicates the stability and fitness of a model, and the closer $R^2$ is to 1, the higher the model fit and the more stable it is [6]. RMSE measures the spread of prediction errors and how concentrated the data are around the line of best fit. RPD is defined as standard deviation (SD) divided by the RMSE, and it is more discriminant than $R^2$ when high $R^2$ is close to 1 [28]. Therefore, a good calibration model is investigated based on the higher $R^2$, the lower RMSE, and higher RPD values.

Williams [29] defined 7 levels of calibration accuracy for NIR systems based on $R^2$ using cereal products, and a value for $R^2$ between 0.50 and 0.65 indicates that more than 50% of the variance in Y is accounted for by the variance in X so that discrimination between high and low concentrations can be made. An $R^2$ value between 0.66 and 0.81 indicates approximate predictions, whereas the value for $R^2$ between 0.82 and 0.90 reveals useful prediction. Calibration models having a value for $R^2$ above 0.91 are considered excellent. Williams [30] reported the RPD statistic and considered 6 levels of prediction accuracy based on the RPD for forages, feeds, and soils samples. For RPD, the calibration is considered very poor and not useful below 1.9. Between 2.0 and 2.4, the calibration is poor and can only be used for rough screening. From 2.5 to 2.9, a fair calibration can be used for screening. Between 3.0 to 3.4, the calibration is good and can be used for quality control. For values between 3.5 and 4.0 and above 4.1, the prediction is classified as very good and excellent, respectively, and the calibration is good for process control and any applications to this type of materials, respectively.

### 3. Results

*3.1. NIR Spectra*

The spectra averaged 100 calibration samples of $NH_4$-N and Org-N spiking groups, respectively, scanned by reflectance and transflectance sensor configurations and are illustrated in Figures 1 and 2. Two large broad peaks at approximately 1450 nm and 1950 nm resulting from the absorption by O-H bonds of water were observed in reflectance HS for both $NH_4$-N and Org-N spiking samples. The reflectance sensor measures a fraction of the incident light that is backscattered at the sample surface and directed to the detection optics, while the transflectance probe is designed for measuring the transmitted part of the incident light through the sample and back to the detector thereafter, and the part of the signal comes from the backscattered light at the sample surface [31]. Generally, the presence of peaks in spectrum indicates a direct correlation between spectral information and analytical concentration, however, the characteristic peaks can be hidden by the overwhelmingly dominant presence of water which makes it difficult to provide simple visual interpretations of the NIR spectra [32]. The NIR predictions of N-related signals mainly originate by the first and second overtones of N-H bonds at 1500–1530 nm and 2168–2180 nm, respectively, and the N-H stretching vibration of a third overtone at 2050–2060 nm. However, the presence of

the N-H bond based on visible peaks is hard to identify in the spectrum (Figures 1 and 2) of this study due to the complete coverage of water absorption. Other interferences can be generally caused by the effects of light scattering and surface roughness of particles in heterogeneous and complex materials such as fresh manure. These interferences can be partially eliminated by choosing appropriate pretreatments of raw spectral data to improve the predictability. Water content is dominant in manure samples; the NIR spectrum that refers to a water reference would improve the signal-to-noise ratio compared to that which refers to a pure white reference, as shown in Figures 1 and 2. The differences between the $NH_4$-N and Org-N spiked samples were not apparent from the sample spectra.

### 3.2. Chemical Composition of Manure Samples

Chemical analyses of the twenty original samples for TS, ash, PS, and manure nutrients including $NH_4$-N, Org-N, P, and K are presented in Table 1. Statistics of the mean, minimum, maximum, and standard deviation (SD) of the calibration dataset are shown in Table 2. As the reliability of the NIR calibration is highly dependent on the data range of constituents, a wide range of dairy cow manure samples with diverse composition is desirable to develop the NIR calibrations [33]. In our case, the TS, ash, and PS ranges varied between 1.4–19.8%, 13.6–50.2%, and 137–628 μm, respectively. The ranges of $NH_4$-N and Org-N were 0.75–4.25 mg mL$^{-1}$ and 0.36–3.75 mg mL$^{-1}$ of the 20 raw manure samples before spiking, respectively. The reliability of the NIR calibration was improved by the chemicals spiking to increase diversity of the N content in the calibration dataset, and they were extended to 17.01 mg mL$^{-1}$ for $NH_4$-N, and 14.28 mg mL$^{-1}$ for Org-N (Table 2). The averaged TS, $NH_4$-N, and Org-N of the raw manure samples in this study were slightly lower compared to those of the previous studies. Reeves and Van Kessel [31] analyzed 107 dairy cow manure samples, and the means of TS, $NH_4$-N, and TN were 11.4% (range: 1.4–38.6%), 1.8 mg mL$^{-1}$ (range: 0.2–4.7 mg mL$^{-1}$), and 4.4 mg mL$^{-1}$ (range: 0.9–9.4 mg mL$^{-1}$), respectively. Finzi et al. [34] conducted a study of using NIR spectroscopy to analyze livestock slurry and digestates and the chemical analysis of 12 dairy slurry showed that the means of TS, $NH_4$-N, and TKN were 9.01% (range: 6.34–11.24%), 1.53 mg mL$^{-1}$ (range: 1.15–2.13 mg mL$^{-1}$), and 3.31 mg mL$^{-1}$ (range: 2.52–4.13 mg mL$^{-1}$), respectively.

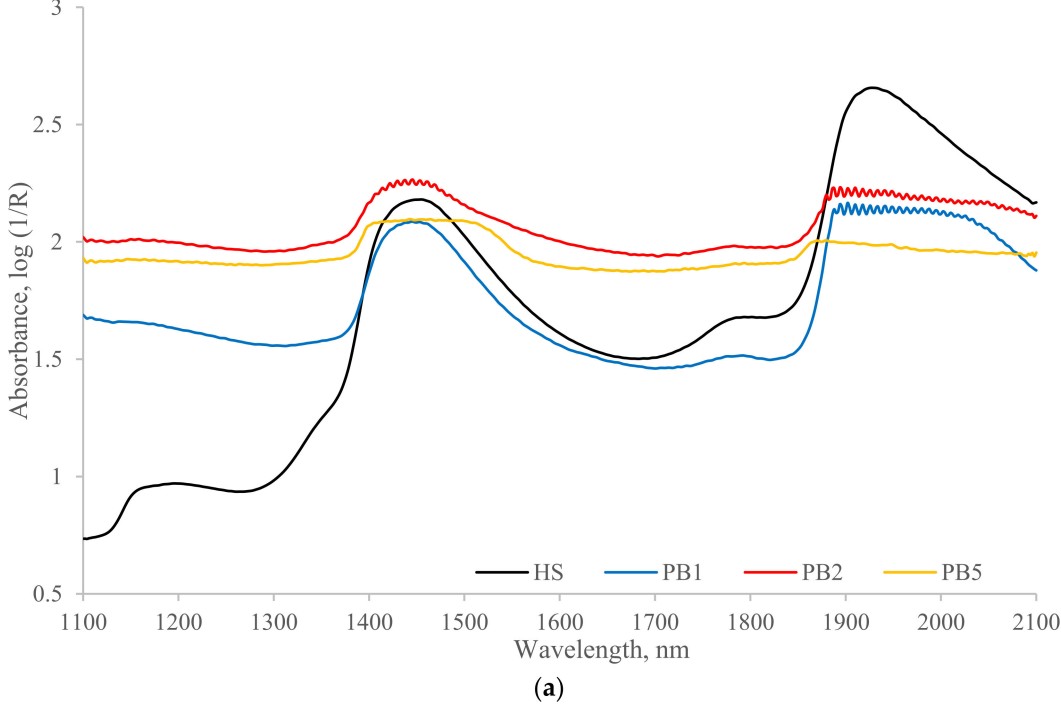

(a)

**Figure 1.** *Cont.*

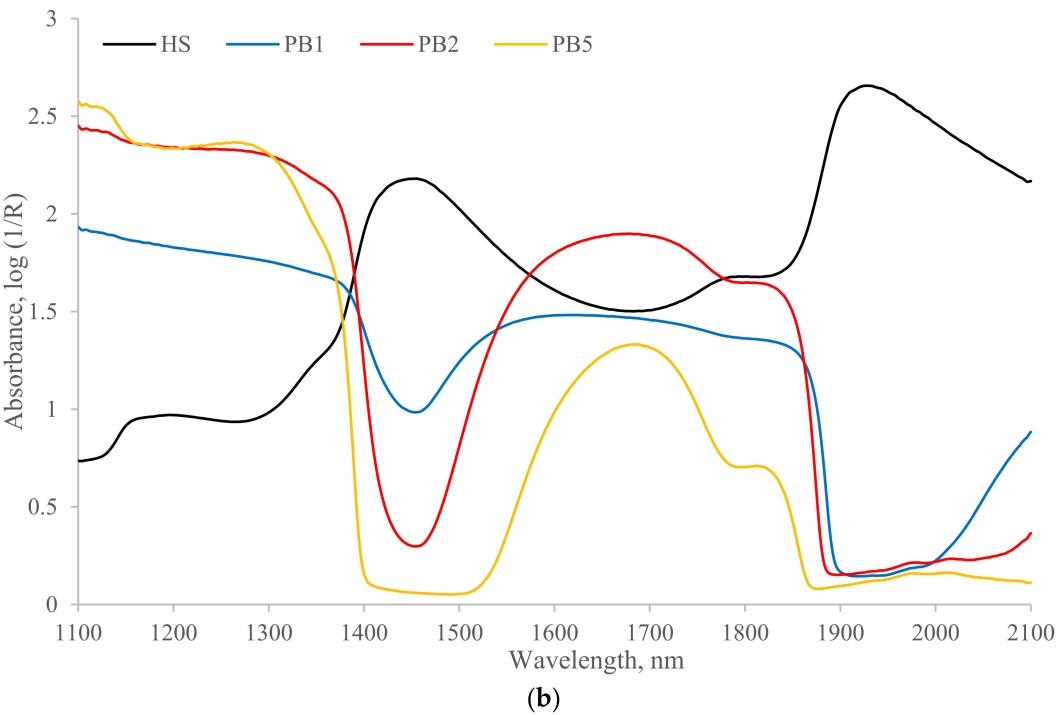

**Figure 1.** (**a**) Averaged spectrum of 100 samples in NH4-N spiking group scanned by one-reflec-tance and three-transflectance sensor configurations. The spectrum of PB1, PB2, and PB5 were converted to pure white reference. (**b**) Averaged spectrum of 100 samples in NH$_4$-N spiking group scanned by one-reflectance and three-transflectance (PB1, PB2, PB5) sensor configurations. The spectrum of PB1, PB2, and PB5 refer to DDI water.

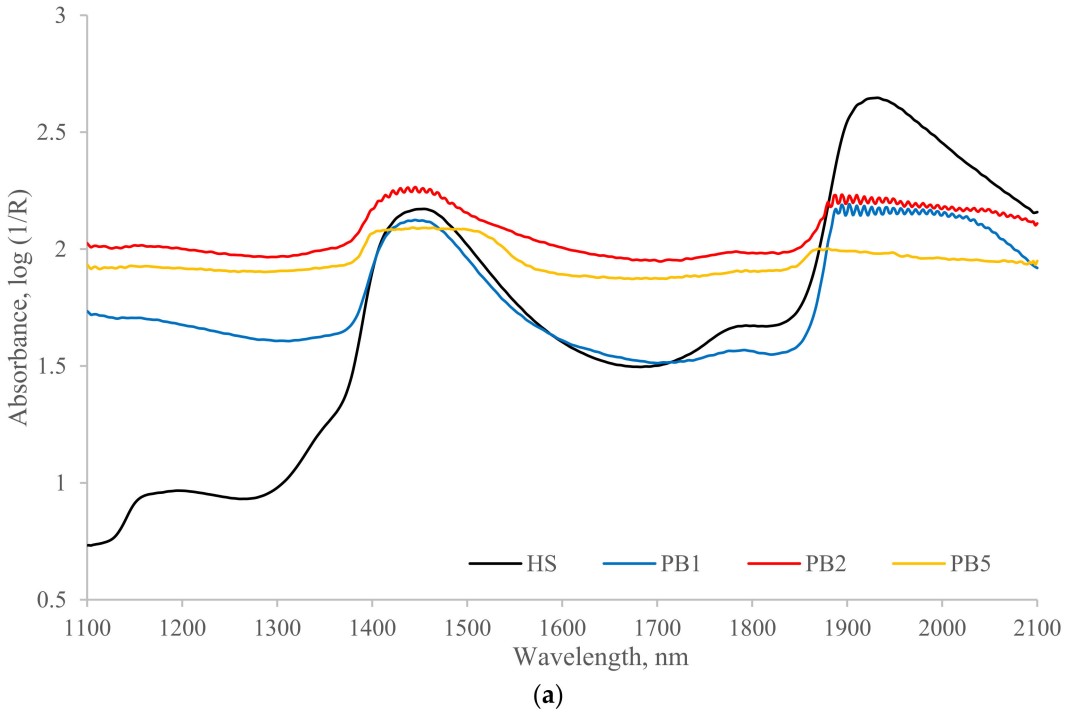

**Figure 2.** *Cont*.

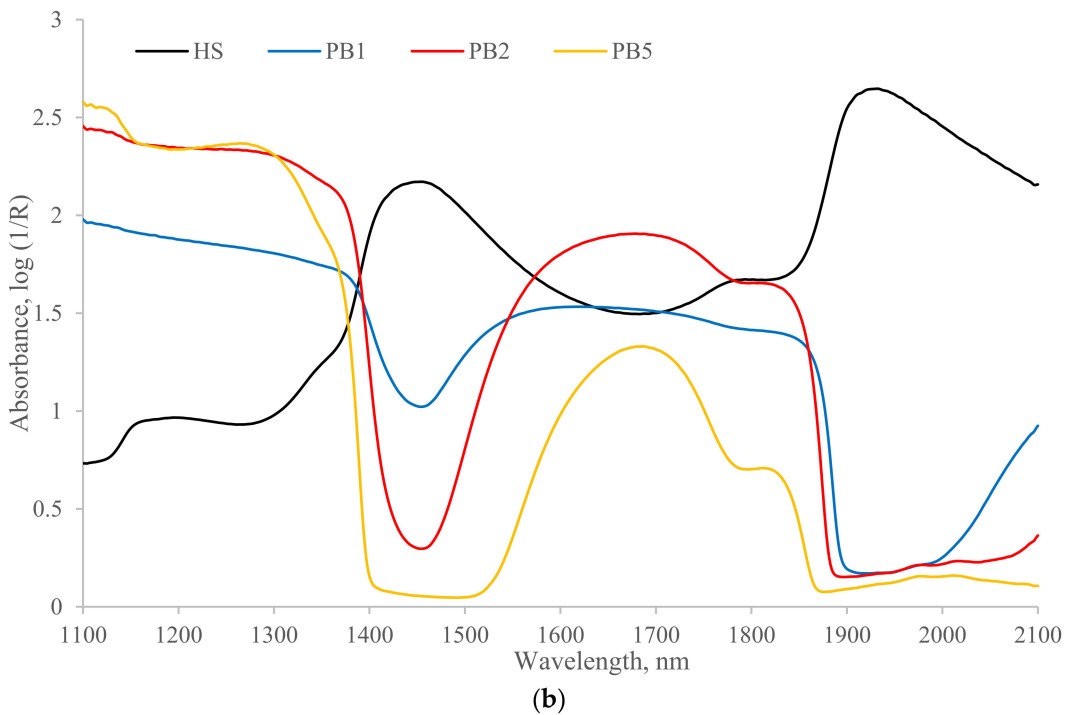

(**b**)

**Figure 2.** (**a**) Averaged spectrum of 100 samples in Org-N spiking group scanned by one-reflec-tance and three-transflectance sensor configurations. The spectrum of PB1, PB2, and PB5 were converted to pure white reference. (**b**) Averaged spectrum of 100 samples in Org-N spiking group scanned by one-reflectance and three-transflectance (PB1, PB2, PB5) sensor configurations. The spectrum of PB1, PB2, and PB5 refer to DDI water.

**Table 1.** Chemical and physical properties of raw manure samples.

| Sample ID | TS (%) | Ash (%) | PS (µM) | NH$_4$-N (mg mL$^{-1}$) | Org-N (mg mL$^{-1}$) | P (mg mL$^{-1}$) | K (mg mL$^{-1}$) |
|---|---|---|---|---|---|---|---|
| S1 | 9.8 | 13.6 | 490 | 1.07 | 1.46 | 0.41 | 1.70 |
| S2 | 8.1 | 14.0 | 498 | 1.13 | 1.37 | 0.42 | 1.56 |
| S3 | 5.4 | 20.6 | 291 | 1.13 | 1.29 | 0.45 | 1.52 |
| S4 | 5.1 | 22.9 | 291 | 1.12 | 1.27 | 0.44 | 1.52 |
| S5 | 3.8 | 28.9 | 347 | 1.13 | 1.25 | 0.43 | 1.40 |
| S6 | 4.7 | 24.6 | 296 | 1.17 | 1.26 | 0.44 | 1.38 |
| S7 | 4.4 | 25.7 | 288 | 1.18 | 1.23 | 0.47 | 1.36 |
| S8 | 3.0 | 28.0 | 278 | 0.96 | 0.98 | 0.36 | 1.30 |
| S9 | 2.8 | 26.6 | 276 | 0.89 | 0.95 | 0.34 | 1.14 |
| S10 | 2.7 | 27.7 | 195 | 0.97 | 0.94 | 0.33 | 1.18 |
| S11 | 19.8 | 50.2 | 343 | 1.50 | 2.84 | 0.81 | 2.00 |
| S12 | 6.1 | 35.8 | 288 | 4.25 | 1.57 | 0.53 | 5.05 |
| S13 | 12.0 | 17.2 | 360 | 1.58 | 2.84 | 0.92 | 3.67 |
| S14 | 4.7 | 21.5 | 314 | 1.01 | 1.09 | 0.47 | 1.85 |
| S15 | 1.4 | 36.4 | 283 | 0.75 | 0.36 | 0.16 | 1.08 |
| S16 | 1.5 | 43.7 | 137 | 0.90 | 0.37 | 0.16 | 1.40 |
| S17 | 8.4 | 41.7 | 314 | 1.34 | 1.42 | 0.61 | 2.20 |
| S18 | 6.3 | 21.1 | 309 | 1.07 | 1.44 | 0.46 | 1.67 |
| S19 | 13.2 | 22.8 | 428 | 1.97 | 2.76 | 0.83 | 2.92 |
| S20 | 13.3 | 17.3 | 628 | 0.93 | 3.57 | 1.10 | 1.49 |

TS = total solid; PS = particle size; NH$_4$-N = ammoniacal nitrogen; Org-N = organic nitrogen; P = phosphorus; K = potassium.

**Table 2.** Statistics of manure compositions of calibration sample set.

| Parameter | TS (%) | Ash (%) | PS ($\mu$M) | NH$_4$-N (mg mL$^{-1}$) | | Org-N (mg mL$^{-1}$) | |
|---|---|---|---|---|---|---|---|
| Spiking | N/A | N/A | N/A | Before | After | Before | After |
| N | 100 | 100 | 100 | 20 | 100 | 20 | 100 |
| Minimum | 1.4 | 13.6 | 137 | 0.75 | 0.75 | 0.36 | 0.36 |
| Mean | 6.8 | 27.0 | 333 | 1.30 | 2.54 | 1.51 | 2.95 |
| Maximum | 19.8 | 50.2 | 628 | 4.25 | 17.01 | 3.57 | 14.28 |
| SD | 4.6 | 9.8 | 107 | 0.75 | 2.16 | 0.84 | 2.46 |

N = sample size; SD = standard deviation; TS = total solid; PS = particle size; NH$_4$-N = ammoniacal nitrogen; Org-N = organic nitrogen.

Common composition of raw excreted dairy cow manure that has not been treated or altered has an average TN content of 4.55–6.47 mg mL$^{-1}$ (37.97–54.01 lbs 1000 gal$^{-1}$) [35]. Utilizing these data as a bounding criterion, the TN content of spiked manure sample that doubled the maximum reported value of 6.47 mg mL$^{-1}$ which is 12.94 mg mL$^{-1}$ would be removed as an outlier. Consequently, S12 spiked for NH$_4$-N at level four, and S19 and S20 spiked for Org-N at level four, were excluded in PLS analysis for N content.

The correlation coefficients (r) between the different constituents of manure samples based on wet chemical measurements in lab were determined (Table 3). Total N in manure is the sum of N forms of NH$_4$-N, Org-N, and nitrate. However, nitrate accounts for a small fraction of TN in manure and is generally negligible and not measured [31]. Higher correlation between Org-N and TN (r = 0.82) was observed and this was because the concentration of Org-N in manure sample was calculated by subtracting the NH$_4$-N from TN. Many studies have reported the correlations between N sources and other manure constituents; however, the results were not consistent and varied significantly among studies. Cabassi et al. [36] reported r greater than 0.65 between ash and DM, Org-N and DM, and Org-N and ash based on analysis of 99 cattle slurry samples. Saeys, Darius, and Ramon [12] reported group-wise correlations among manure constituents and divided the constituents into a DM-group and a N-group including the parameters that were highly correlated with DM content and TN content, respectively. According to Saeys' analyses, NH$_4$-N and K were highly correlated to TN content and were assigned to the N-group, and this was comparable to the results in this study [12,33]. The r between TN and P was 0.80 which indicated a high correlation, and this was not observed in previous research.

**Table 3.** Correlation coefficients (r) among the chemical constituents in raw samples of dairy cow manure (N = 20).

| Parameter | TS | Ash | PS | NH$_4$-N | Org-N | TN | P | K |
|---|---|---|---|---|---|---|---|---|
| TS | 1.00 | | | | | | | |
| Ash | −0.32 | 1.00 | | | | | | |
| PS | 0.17 | −0.62 | 1.00 | | | | | |
| NH$_4$-N | 0.18 | 0.16 | 0.00 | 1.00 | | | | |
| Org-N | −0.17 | −0.34 | 0.70 | 0.25 | 1.00 | | | |
| TN | −0.01 | −0.13 | 0.46 | 0.76 | 0.82 | 1.00 | | |
| P | −0.14 | −0.31 | 0.65 | 0.25 | 0.98 | 0.80 | 1.00 | |
| K | 0.07 | −0.01 | 0.09 | 0.91 | 0.41 | 0.82 | 0.44 | 1.00 |

TS = total solid; PS = particle size; NH$_4$-N = ammoniacal nitrogen; Org-N = organic nitrogen; TN = total nitrogen; P = phosphorus; K = potassium.

### 3.3. Constituent PLS Calibrations of Reflectance and Transflectance Sensors

The best cross-validation results for manure constituents in reflectance and transflectance modes are presented in Table 4 for NH$_4$-N spiking group and Table 5 for Org-N spiking group. The scatter plots and linear regression between NIR predictions and reference values are shown in Figures 3 and 4.

**Table 4.** NIR calibrations (*n* = 100) of manure constituents using reflectance and transflectance sensors based on the $NH_4$-N spiking sample group.

| Configuration | Parameter | Pretreatment | Outliers | LVs | $R^2$ (CV) | RMSE (CV) | RPD (CV) |
|---|---|---|---|---|---|---|---|
| HS | $NH_4$-N | MC; D-1,1,15 | 1 | 7 | 0.83 | 0.65 | 2.45 |
| | TS | MSC; D-1,1,15 | 0 | 7 | 0.90 | 0.015 | 3.16 |
| | Ash | MC; D-1,2,15 | 0 | 7 | 0.66 | 0.057 | 1.71 |
| | PS | D-2,2,15 | 0 | 5 | 0.69 | 60.0 | 1.79 |
| PB1 | $NH_4$-N | D-1,2,15 | 1 | 9 | 0.56 | 1.07 | 1.50 |
| | TS | D-1,1,15 | 0 | 8 | 0.97 | 0.008 | 5.50 |
| | Ash | MC; D-2,2,15 | 0 | 7 | 0.86 | 0.037 | 2.64 |
| | PS | MC; D-1,2,15 | 0 | 8 | 0.78 | 49.4 | 2.15 |
| PB2 | $NH_4$-N | MSC; D-1,2,15 | 1 | 1 | 0.30 | 1.64 | 1.20 |
| | TS | MC | 0 | 8 | 0.88 | 0.016 | 2.86 |
| | Ash | MSC | 0 | 8 | 0.90 | 0.031 | 3.10 |
| | PS | MSC | 0 | 5 | 0.70 | 58.5 | 1.81 |
| PB5 | $NH_4$-N | MSC; D-2,2,15 | 1 | 1 | 0.37 | 1.65 | 1.26 |
| | TS | MSC | 0 | 8 | 0.89 | 0.015 | 3.03 |
| | Ash | MSC | 0 | 9 | 0.89 | 0.032 | 3.00 |
| | PS | MSC; D-1,2,15 | 0 | 8 | 0.74 | 53.9 | 1.97 |

HS = reflectance head sensor; PB = transflectance probe with optical path length of 1 mm, 2 mm, and 5 mm; TS = total solid; PS = particle size; $NH_4$-N = ammoniacal nitrogen; LV = latent variable; CV = cross-validation; RMSE = root mean square error, and the units of $NH_4$-N, TS, Ash, and PS are mg mL$^{-1}$, %, %, and μM, respectively; RPD = residual prediction deviation; MC = mean center; MSC = multiplicative scatter correction; D = Savitzky-Golay derivative: first or second derivative, first or quadratic polynomial interpolation, smoothing kernel of length 15.

**Table 5.** NIR calibrations (*n* = 100) of manure constituents using reflectance and transflectance sensors based on the Org-N spiking sample group.

| Configuration | Parameter | Pretreatment | Outliers | LVs | $R^2$ (CV) | RMSE (CV) | RPD (CV) |
|---|---|---|---|---|---|---|---|
| HS | Org-N | D-1,2,15 | 2 | 12 | 0.66 | 1.18 | 1.73 |
| | TS | MC; D-1,2,15 | 0 | 9 | 0.90 | 0.015 | 3.16 |
| | Ash | MC; D-1,1,15 | 0 | 9 | 0.72 | 0.053 | 1.88 |
| | PS | MC; D-2,2,15 | 0 | 5 | 0.67 | 61.9 | 1.73 |
| PB1 | Org-N | MC; D-1,1,15 | 2 | 6 | 0.34 | 1.67 | 1.23 |
| | TS | MC; D-1,1,15 | 0 | 6 | 0.97 | 0.009 | 5.42 |
| | Ash | MC; D-1,1,15 | 0 | 6 | 0.87 | 0.035 | 2.77 |
| | PS | MSC; D-1,1,15 | 0 | 7 | 0.77 | 50.7 | 2.09 |
| PB2 | Org-N | MC; D-1,2,15 | 2 | 8 | 0.27 | 1.83 | 1.17 |
| | TS | MC | 0 | 8 | 0.92 | 0.013 | 3.58 |
| | Ash | MSC | 0 | 8 | 0.88 | 0.034 | 2.86 |
| | PS | MSC | 0 | 8 | 0.78 | 50.4 | 2.11 |
| PB5 | Org-N | MSC; D-1,2,15 | 2 | 3 | 0.15 | 1.89 | 1.08 |
| | TS | MSC | 0 | 7 | 0.80 | 0.021 | 2.26 |
| | Ash | MSC; D-1,2,15 | 0 | 8 | 0.85 | 0.038 | 2.56 |
| | PS | MC | 0 | 7 | 0.77 | 51.6 | 2.07 |

HS = reflectance head sensor; PB = transflectance probe with optical path length of 1 mm, 2 mm, and 5 mm; TS = total solid; PS = particle size; Org-N = organic nitrogen; LV = latent variable; CV = cross-validation; RMSE = root mean square error, and the units of Org-N, TS, Ash, and PS are mg mL$^{-1}$, %, %, and μM, respectively; RPD = residual prediction deviation; MC = mean center; MSC = multiplicative scatter correction; D = Savitzky-Golay derivative: first or second derivative, first or quadratic polynomial interpolation, smoothing kernel of length 15.

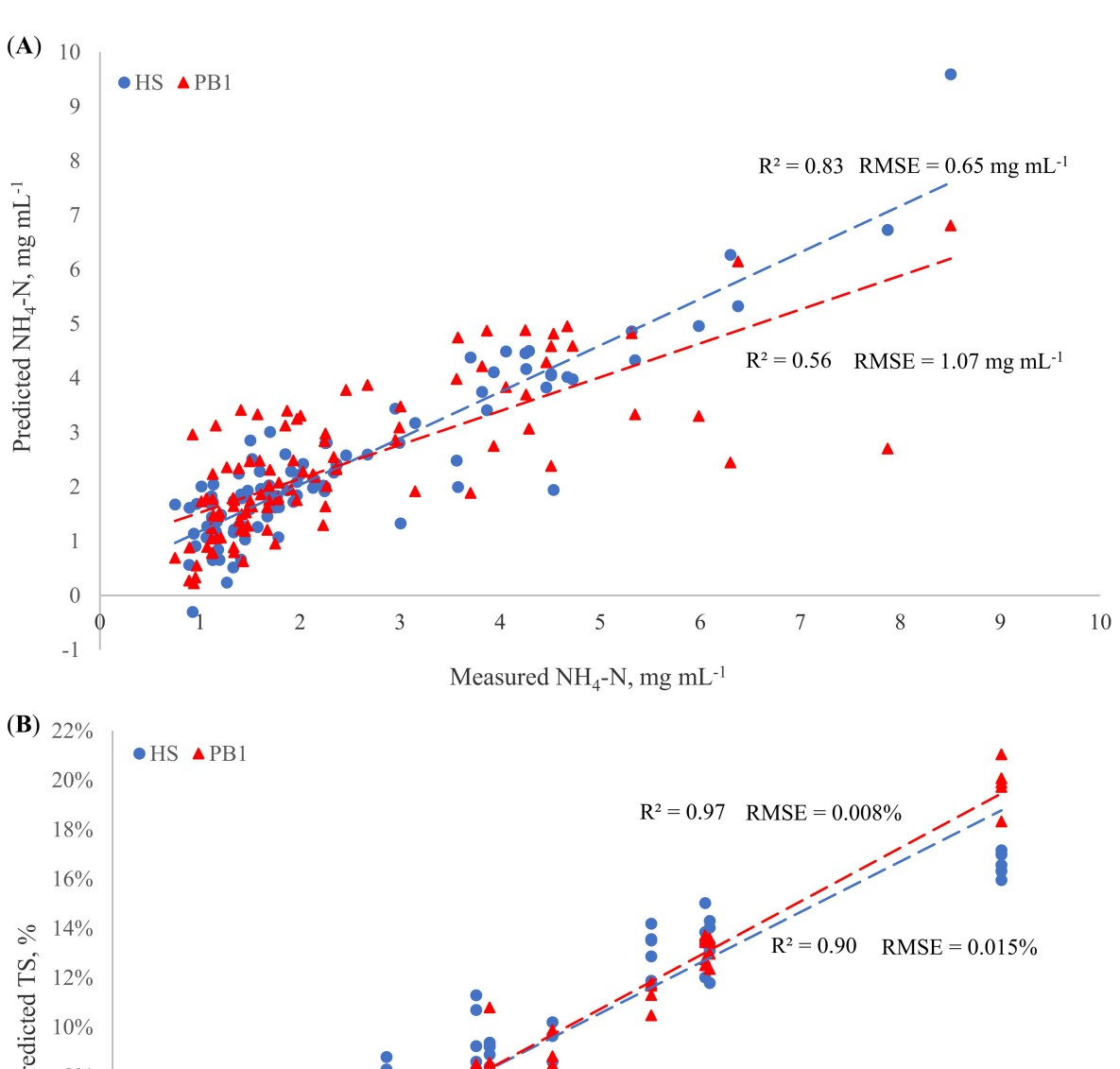

**Figure 3.** *Cont.*

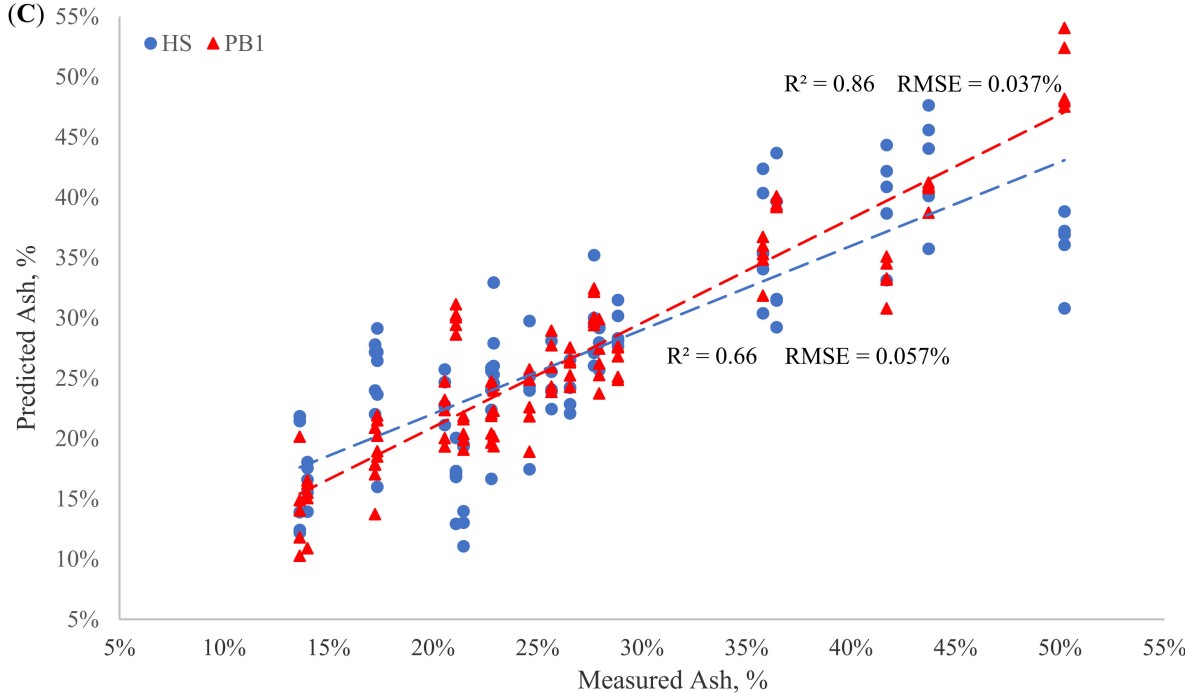

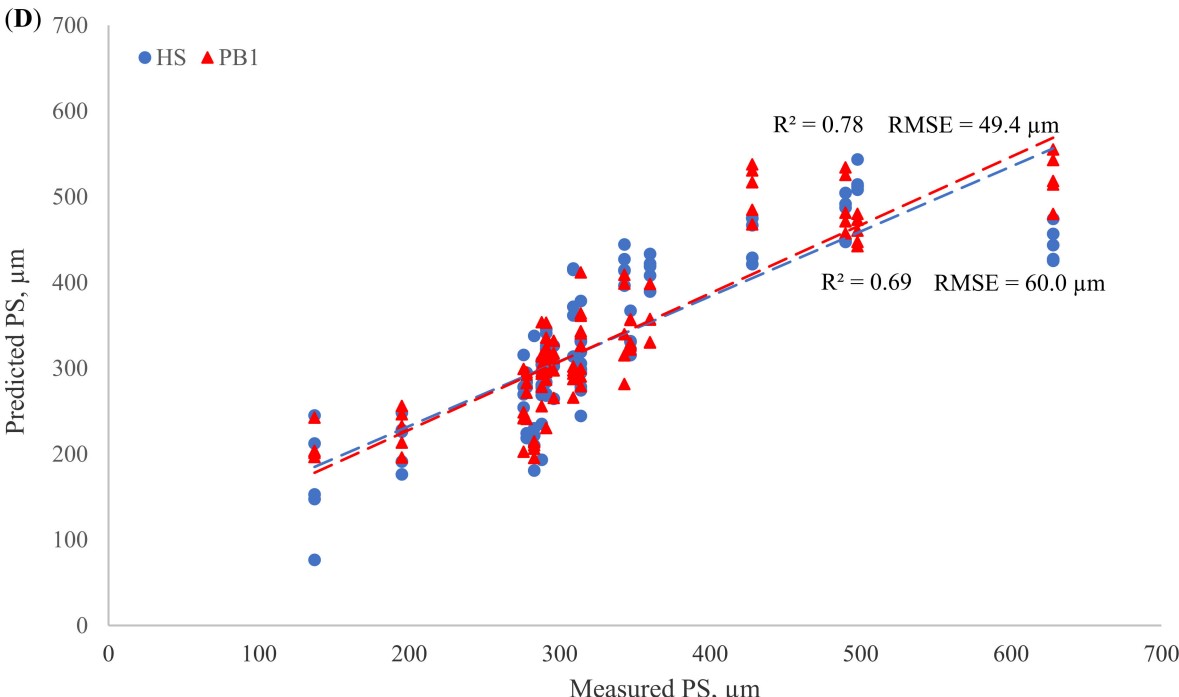

**Figure 3.** The PLS linear relationships between NIR predictions and lab measurements for manure constituents using reflectance (HS) and transflectance (PB1) sensors based on NH$_4$-N spiking group: (**A**) NH$_4$-N content; (**B**) total solid (TS); (**C**) ash content; (**D**) particle size (PS).

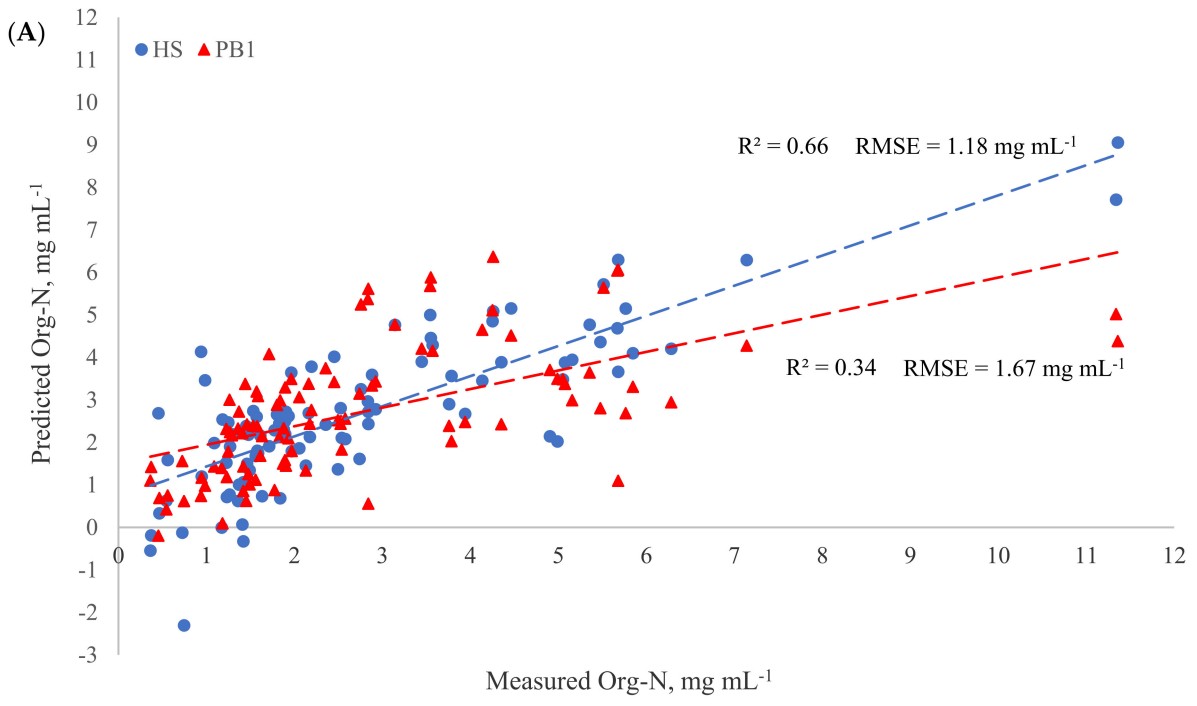

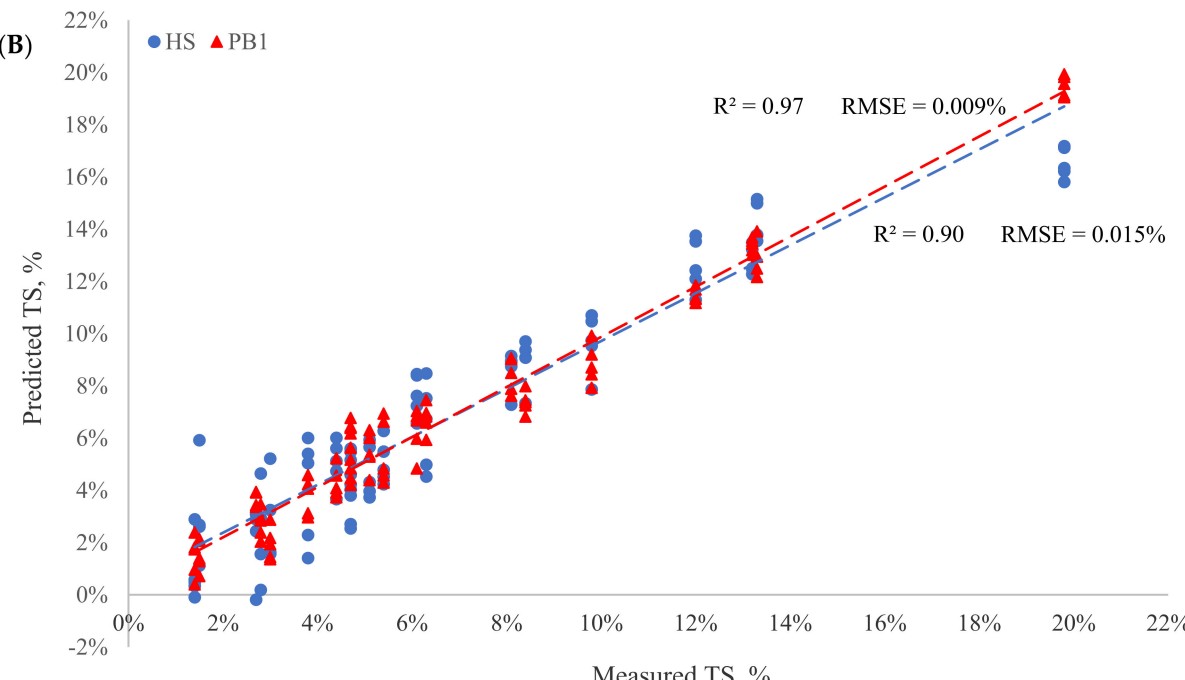

**Figure 4.** *Cont.*

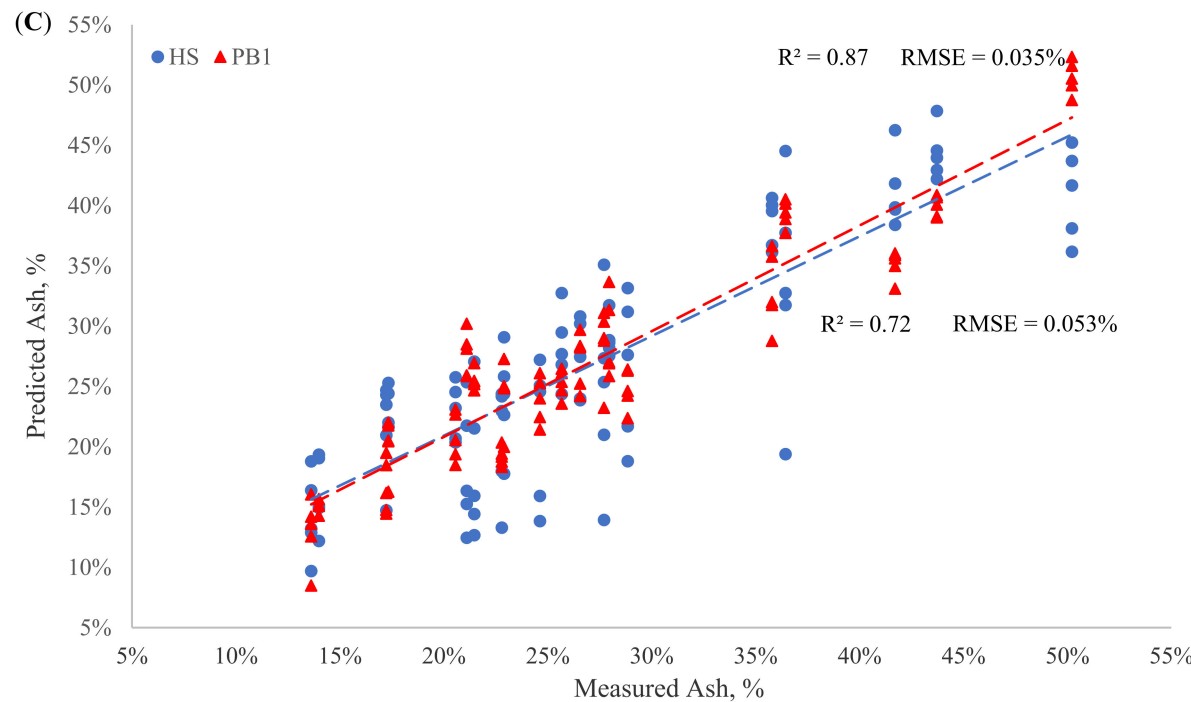

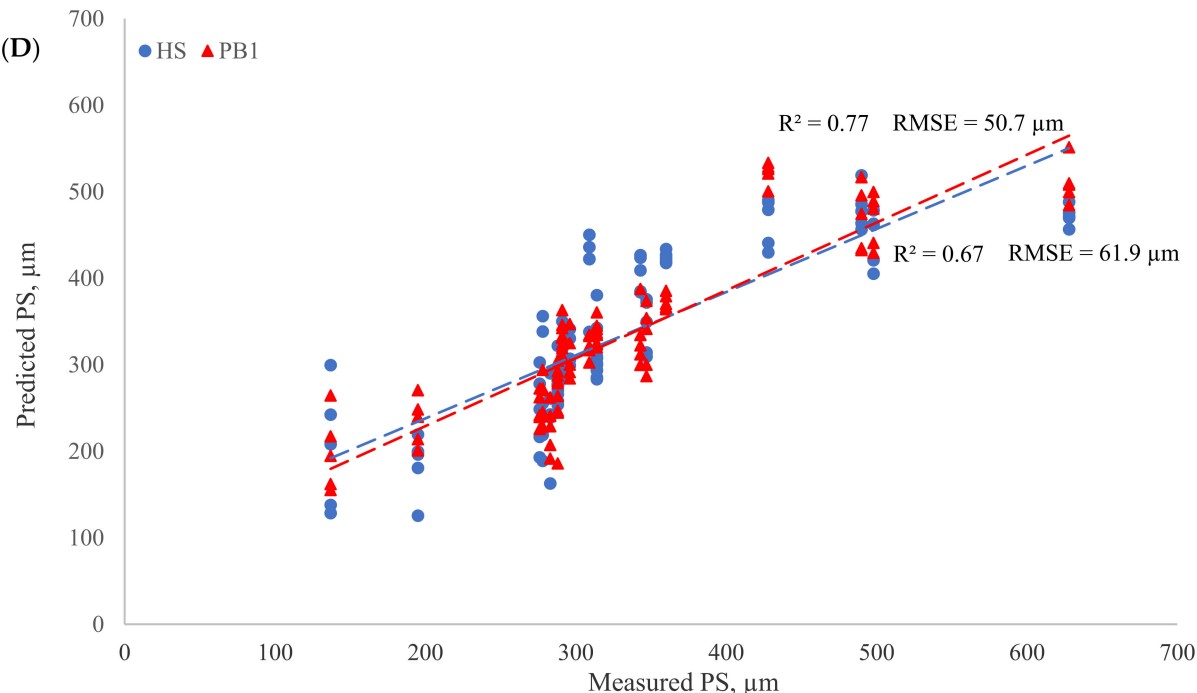

**Figure 4.** The PLS linear relationships between NIR predictions and lab measurements for manure constituents using reflectance (HS) and transflectance (PB1) sensors based on Org-N spiking group: (**A**) Org-N content; (**B**) total solid (TS); (**C**) ash content; (**D**) particle size (PS).

### 3.3.1. Path Lengths

Among the three transflectance configurations, the PB1 configuration was the best for predicting $NH_4$-N content, with an $R^2$ of 0.56 and an RMSE of 1.07 mg mL$^{-1}$. However, based on the established criteria, this calibration with an RPD of 1.50 would only have utility to separate high and low values (Table 4). The calibration accuracy of the PB2 and PB5 for $NH_4$-N content were similar (PB2: $R^2$ = 0.30, RMSE = 1.64 mg mL$^{-1}$; PB5: $R^2$ = 0.37, RMSE = 1.65 mg mL$^{-1}$) and the low $R^2$ values indicated PB2 and PB5 were

not useful for predicting the concentrations of $NH_4$-N. The number of LVs for PB2 and PB5 for $NH_4$-N was only one, which indicated that the regression should be regarded with caution because based on the concepts of PLS analysis, it was unlikely that all the variance in spectral and reference data was explicable by a single factor [27]. The $R^2$ for TS achieved with spectra collected in the transflectance mode were all greater than 0.88 which revealed good-to-excellent calibration accuracy. Specifically, comparing predicted TS to laboratory reference values yielded an $R^2$ of 0.97 and an RMSE of 0.008%, indicating excellent quantitative predication of PB1 (RPD = 5.5). The TS predicted by PB2 and PB5 had an $R^2$ of 0.88 and 0.89 and RMSE of 0.016% and 0.015%, respectively, and RPD values of 2.86 and 3.03 showed fair-to-good classifications according to the established criteria. For predicting ash content, the PB2 configuration had the best performance with an $R^2$ of 0.90 and an RMSE of 0.031%. The PB1 and PB5 configurations had $R^2$ of 0.86 and 0.89 and RMSE of 0.037% to 0.032%, respectively, and both indicated good calibration accuracy (Table 4). The results of $R^2$ were in the range of 0.70–0.78 with RMSE of 49.4–58.5 μm for PS and presented approximate accuracy of all three transflectance configurations, and the RPDs were between 1.81 and 2.15 which revealed the prediction accuracy based on RPD statistic was poor (Table 4). Overall, the transflectance configuration of PB1 presented the best predictions for $NH_4$-N, TS, and PS among the three optical path lengths. For ash content, the $R^2$ and RMSE of PB2 were slightly better than those of PB1 and PB5, however, the performance of the three configurations for ash calibrations were similar.

The predictions of manure constituents with different optical path lengths in transflectance mode based on the Org-N spiking group are shown in Table 5. Prediction models based on the transflectance configurations were only able to account for 34% of the variance in predicted Org-N content. The calibration accuracy of TS decreased as the optical path length of the transflectance probe increased from 1 mm to 5 mm. The TS predicted by PB1 was excellent with an $R^2$ and RMSE of 0.97 and 0.009%, respectively, and an RPD of 5.42 illustrated the calibration was usable in any application. Similar to the results in the $NH_4$-N spiking group, PB2 showed better performance for predicting ash content with an $R^2$ of 0.88 and an RMSE of 0.034% compared to the PB1 and PB5 configurations. For predicting PS, the results of the three configurations were similar and had no significant differences.

The results of NIR predictions in transflectance mode for TS, ash, and PS between the $NH_4$-N and Org-N spiking groups indicated that these manure constituents could be predicted independent of the spectral influence of these chemical species.

### 3.3.2. Reflectance and Transflectance PLS Calibration

To investigate the performance of reflectance and transflectance sensors, linear regression was used to evaluate NIR predictions versus lab measurements for manure constituents using reflectance HS and transflectance PB1 based on $NH_4$-N spiking group (Figure 3, Table 4). The $NH_4$-N content predicted by NIR had an $R^2$ of 0.83 and an RMSE of 0.65 mg mL$^{-1}$ for reflectance HS, and an $R^2$ of 0.56 and an RMSE of 1.07 mg mL$^{-1}$ for transflectance PB1, respectively (Table 4). The reflectance HS was useful in predicting $NH_4$-N content and performed better than the transflectance PB1 which could only discriminate the high and low concentrations of $NH_4$-N in manure. However, the $NH_4$-N predicted by reflectance HS (RPD = 2.45) and transflectance PB1 (RPD = 1.50) were classified as poor and fair based on the RPD statistic. Figure 4 and Table 5 show the linear regression results of NIR predictions versus lab measurements for manure constituents using reflectance and transflectance sensors based on the Org-N spiking group. The Org-N content predicted by the reflectance HS with $R^2$ of 0.66 and RMSE of 1.18 mg mL$^{-1}$ was more accurate than the transflectance PB1 with $R^2$ of 0.34 and RMSE of 1.67 mg mL$^{-1}$ (Table 5). The reflectance HS approximately predicted Org-N content, whereas the transflectance PB1 was not feasible for Org-N prediction as the $R^2$ was under 0.5. The number of LVs for the HS Org-N model was 12 which was relatively large, and this might indicate some uncertainty in the model computation [27].

The $R^2$ and RMSE of transflectance PB1 were 0.97 and 0.008% in the $NH_4$-N spiking group, respectively (Table 4), which indicated that the accuracy of TS predictions was excellent, and the calibration was usable in any applications (RPD = 5.50). Although TS predicted by the reflectance HS was less accurate than the PB1, it provided good quantitative prediction with an $R^2$ of 0.90 and an RMSE of 0.015%. The HS predicted ash and PS approximately; the $R^2$ and RMSE were 0.66 and 0.057% for ash, and 0.69 and 60.0 μm for PS, respectively (Table 4). The $R^2$ and RMSE values of 0.86 and 0.037% for ash content and 0.78 and 49.4 μm for PS, respectively, indicated the transflectance PB1 was better than the reflectance HS in predicting ash and PS in manure samples of the $NH_4$-N spiking group (Table 4). The calibration evaluations of reflectance and transflectance modes for TS, ash, and PS in the Org-N spiking group were all similar to the results in the $NH_4$-N spiking group that the transflectance PB1 (TS: $R^2$ = 0.97, RMSE = 0.009%; ash: $R^2$ = 0.87, RMSE = 0.035%; PS: $R^2$ = 0.77, RMSE = 50.7 μm) presented a better performance than the reflectance HS (TS: $R^2$ = 0.90, RMSE = 0.015%; ash: $R^2$ = 0.72, RMSE = 0.053%; PS: $R^2$ = 0.67, RMSE = 61.9 μm) for predicting TS, ash, and PS, respectively (Table 5).

Therefore, the reflectance HS had a better performance than the transflectance probe for predicting $NH_4$-N and Org-N concentrations. Both the reflectance and transflectance provided excellent calibration accuracy for TS, however the transflectance probe was more accurate than the reflectance sensor. The transflectance probe had a better performance than the reflectance sensor for predicting ash content and PS.

### 3.4. Relationship between Nitrogen and Other Manure Parameters

To investigate whether the NIR predictions of N sources were affected by the correlation between the N content and manure characteristics, or directly predicted by the changes in N concentrations, each sample with its corresponding five spiking levels was grouped together and labeled from $SG_{NH4N}1$ to $SG_{NH4N}20$ for $NH_4$-N spiking group (SG) samples, and from $SG_{OrgN}1$ to $SG_{OrgN}20$ for Org-N spiking group samples. The calibration dataset was distributed into calibration and validation parts, one SG sample was set as the validation dataset, and the other 19 SGs (with their corresponded spiking levels) were used for the PLS calibration dataset. This procedure was repeated 20 times, setting each SG as a validation dataset, and therefore twenty $R^2_{SG}$ of linear regression between the predicted versus measured N content of SG were calculated.

The linear regressions between the twenty $R^2_{SG}$ of $NH_4$-N predictions and the measured TS, ash content, and PS of the predicted sample, respectively, using reflectance HS and transflectance PB1 are summarized in Table 6. For each $SG_{NH4-N}$, the manure constituents including TS, ash, and PS were the same for the five spiked samples, and the concentration of $NH_4$-N was the only variance, and thus a high $R^2_{SG}$ indicated the $NH_4$-N content was accurately predicted by the NIR spectroscopy independent of TS, ash content, and PS of the manure. Although most of the $R^2_{SG}$ were greater than 0.8 indicating good predictions of $NH_4$-N within each spiking group, the prediction accuracy of $NH_4$-N for individual SG was not consistent and varied from 0.014 to 0.992 for reflectance HS and from 0.000 to 0.998 for transflectance PB1, respectively (Figure S1). Such low accuracies in predicting $NH_4$-N in some SG samples may relate to the small sample size ($n$ = 5), with a relatively large SD of each SG. The coefficients of determination $R^2_r$ between the $R^2_{SG}$ and TS were 0.14 for HS and 0.30 for PB1 (Table 6) which indicated a weak correlation between the accuracy of $NH_4$-N prediction and manure TS content for both reflectance and transflectance sensors. Similarly, ash content and PS of manure had little effect on $NH_4$-N predictions for both HS and PB1 (Table 6).

**Table 6.** Linear regression of the twenty $R^2_{SG}$ for predicting $NH_4$-N and Org-N, respectively, versus the total solid (TS), ash content, and particle size (PS) of the predicted manure sample, respectively.

| Parameter | $NH_4$-N | | | Org-N | | |
|---|---|---|---|---|---|---|
| | Sensor | Equation | $R_r^2$ | Sensor | Equation | $R_r^2$ |
| TS | HS | 1.89x + 0.69 | 0.14 | HS | 2.71x + 0.55 | 0.27 |
| | PB1 | −3.33x + 1.03 | 0.30 | PB1 | −1.81x + 0.77 | 0.10 |
| Ash | HS | −0.28x + 0.90 | 0.01 | HS | −1.20x + 1.05 | 0.24 |
| | PB1 | 0.42x + 0.69 | 0.02 | PB1 | −1.30x + 1.00 | 0.22 |
| PS | HS | 0.00x + 0.80 | 0.00 | HS | 0.00x + 0.22 | 0.47 |
| | PB1 | −0.00x + 1.10 | 0.12 | PB1 | 0.00x + 0.56 | 0.01 |

SG = spiking group; $R^2_r$ = coefficients of determination of $R^2_{SG}$ vs. manure constituents; $R^2_{SG}$ = the $R^2$ of PLS calibration using 19 SG samples as calibration and 1 SG sample as validation dataset; HS = reflectance head sensor; PB1 = transflectance probe with optical path length of 1 mm; TS = total solid; PS = particle size; $NH_4$-N = ammoniacal nitrogen; Org-N = organic nitrogen.

The results of the $R^2_{SG}$ of individual SG for predicting Org-N versus other manure compositions are presented in Table 6. The $R^2_{SG}$ was between 0.149 and 0.988 for reflectance HS and between 0.149 and 0.996 for transflectance PB1, respectively (Figure S2). Strong linear patterns between the $R^2_{SG}$ for predicting Org-N and TS and ash content of manure samples, respectively, were not observed for both HS and PB1(Table 6). The $R^2_r$ values of $R^2_{SG}$ versus PS for HS and PB1 were 0.47 and 0.01, respectively (Table 6), which indicated the PS had a greater effect on the predicted accuracy of Org-N content for reflectance HS than for transflectance PB1.

The non-linear relationships between the $R^2_{SG}$ of $NH_4$-N and Org-N predictions and TS, ash, and PS of manure samples show that the NIR spectra can predict the concentrations of $NH_4$-N and Org-N in manure independent of TS, ash content, and PS of the manure samples. This result indicates that the NIR spectrum is influenced directly by the N source, and prediction models for $NH_4$-N and Org-N can be made independent of correlations between the N source and other manure constituent values.

## 4. Discussion

### 4.1. Effect of Path Lengths of Transflectance Mode

The performances of the transflectance probe with three configurations of different optical path lengths were evaluated for manure samples of both $NH_4$-N and Org-N spiking groups. The averaged raw spectra of transflectance configurations PB1 and PB2 showed high noise level at 1900–2100 nm which indicated most light was blocked and that the NIR measurement was not accurate beyond 1900 nm (Figures 1a and 2a). Moreover, the noise level of PB2 was higher than the PB1 at approximately 1450 nm. The light was completed blocked and broad peaks at 1450 nm and 1950 nm were not observed in the spectrum of PB5 in Figures 1 and 2, which indicated the absorbance measured by PB5 was not accurate. According to the NIR calibration results, the transflectance probe with the shortest optical path length (PB1) presented the most accurate predictions of manure constituents, including $NH_4$-N, Org-N, and TS, among the three path lengths (Tables 4 and 5). The performances of PB1 for predicting ash and PS were similar to PB2 and PB5. An explanation of this result could be that with a longer path length, the light travels through more solution and is absorbed, and this would increase the absorbance and have less light transmit through the sample. Consequently, the spectrum of PB1 configuration which has the shortest pathway is less saturated compared to PB2 and PB5 with longer pathways during the NIR measurements, resulting in a better predicted utility.

### 4.2. NIR Prediction for Nitrogen Concentrations

The PLS calibration results of reflectance and transflectance modes indicated that the reflectance HS provided better predictions for $NH_4$-N and Org-N than the transflectance probe. The calibration of the $NH_4$-N content using NIR reflectance was useful, and the

reflectance HS provided approximately quantitative predictions of Org-N. The NIR transflectance only separated high and low concentrations of $NH_4$-N, whereas it was not usable for predicting Org-N. The performance of the NIR technique for analyzing $NH_4$-N content has been investigated in many previous studies and good quantitative predictions in different animal specials have been presented. However, only a few studies have investigated the use of NIR with both transflectance and reflectance sensors analyzing manure nutrients. Reeves and Van Kessel [37] calibrated the NIR reflectance model using 107 dairy cow's manure and the $R^2$ for $NH_4$-N prediction was 0.83, which agree to the results in this study. Saeys et al. [16] used reflectance and transflectance modes of NIR spectroscopy to analyze $NH_4$-N content in swine manure and the $R^2$ and RPD were 0.77 and 2.10 using reflectance mode and 0.76 and 2.06 using transflectance mode, respectively, and our results partially agreed with those findings that the reflectance presented better predictions for $NH_4$-N than the transflectance mode. However, the $R^2$ and RPD of reflectance was significantly higher ($p < 0.05$) than those of transflectance in this study, whereas the calibrations for $NH_4$-N of the two modes were close, as reported by Saeys et al. [16]. This may be explained by the different animal species of manure. The calibrations for Org-N using both reflectance and transflectance were less accurate than those for $NH_4$-N content. The calibration of Org-N content in this study was less accurate compared to the previous analysis. Based on the NIR reflectance spectra of 135 composted swine manure samples, Nam and Lee [9] reported that the $R^2$ for the Org-N calibration model was 0.76. Reeves [11] analyzed 207 poultry manure samples using NIR and the $R^2$ was 0.89 for Org-N.

*4.3. Future Work*

Future work may be required to improve the accuracy and precision of calibration models; validate the calibration models using external manure samples within the prediction range of N from other dairy cow farms; enhance the procedure of spiking method in manure analysis using NIR techniques; investigate the methodology with other manure constituents such as P and K; and conduct online variable rate application of organic fertilizer using NIR sensing system.

**5. Conclusions**

This study measured the spectra of dairy cow manure samples using a NIR spectroscopy in both reflectance and transflectance modes. Manure samples were collected from six dairy cow farms and spiked with chemical compounds to alter $NH_4$-N and Org-N contents in the calibration dataset. The NIR models for predicting the constituents in dairy cow manure were developed using the spiked samples and the performance of the NIR predictions of different sensor configurations were evaluated. The relationship of NIR predictions for N contents independent of other manure physical and chemical properties was explored in this study. The results illustrated a transflectance probe with 1 mm optical path length (PB1) showed the best performance for predicting $NH_4$-N, Org-N, and TS. The differences of calibration performance for ash and PS among three transflectance configurations (PB1, PB2, PB5) were small. To compare the reflectance and transflectance modes of the NIR system, the transflectance probe (PB1) yielded prediction models with better performance for TS, ash, and PS when evaluating $R^2$ and RMSE of both spiking groups. Calibration models of the reflectance sensor (HS) had better performance than the transflectance probe (PB1) for predicting $NH_4$-N (HS: $R^2$ = 0.83, RMSE = 0.65 mg $mL^{-1}$; PB1: $R^2$ = 0.56, RMSE = 1.07 mg $mL^{-1}$) and Org-N (HS: $R^2$ = 0.66, RMSE = 1.18 mg $mL^{-1}$; PB1: $R^2$ = 0.34, RMSE = 1.67 mg $mL^{-1}$) concentrations with higher $R^2$s and lower RMSEs. Additionally, the HS models provided good prediction for $NH_4$-N ($R^2$ = 0.83) and approximate predictions for Org-N ($R^2$ = 0.66); however, transflectance PB1 was able to discriminate high and low values for $NH_4$-N ($R^2$ = 0.56) and was not feasible to use for predicting Org-N ($R^2$ = 0.34). The calibration of TS using transflectance PB1 was excellent with an $R^2$ of 0.97 and was usable for any applications. Transflectance PB1 provided good quantitative predictions for ash and approximate predictions for PS. The correlations be-

tween the accuracy of NIR predictions for $NH_4$-N and Org-N concentrations and TS, ash, and PS of dairy cow manure were not observed.

In summary, the transflectance was more accurate than the reflectance for predicting TS, ash, and PS, while the reflectance provided better predictions of N speciation than the transflectance in dairy cow manure using the NIR system. The NIR sensors can predict the N concentrations without being affected by other manure characteristics of TS, ash content, and PS. The results obtained in this study indicate that the spiking method of adding chemical N sources to provide adequate calibration samples has the potential to conduct a rapid and cost-effective analysis of dairy cow manure, including $NH_4$-N, Org-N, TS, ash, and PS, using NIR spectroscopy.

**Supplementary Materials:** The following supporting information can be downloaded at: https://www.mdpi.com/article/10.3390/rs14040963/s1, Figure S1: Linear regression of the $R^2_{SG}$ of induvial spiking group (SG) for predicting $NH_4$-N vs. other manure compositions; Figure S2: Linear regression of the $R^2_{SG}$ of induvial spiking group (SG) for predicting Org-N vs. other manure compositions.

**Author Contributions:** Conception, designation, methodology: R.A.L., M.F.D. and X.F. Laboratory measurements, data analysis, original draft writing: X.F. Funding acquisition, project administration, review and editing: R.A.L. and M.F.D. All authors revised the paper. All authors have read and agreed to the published version of the manuscript.

**Funding:** Support for this project was provided by the Wisconsin Dairy Innovation Hub.

**Institutional Review Board Statement:** Not applicable.

**Informed Consent Statement:** Not applicable.

**Data Availability Statement:** Not applicable.

**Conflicts of Interest:** The authors declare no conflict of interest.

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
