# Peer review of "Evaluation of Near-Infrared Reflectance and Transflectance Sensing System for Predicting Manure Nutrients"

_remotesensing, doi:10.3390/rs14040963_

Round 1

Reviewer 1 Report

Specific comments:

Abstract:

The authors included the errors (RMSE), but they did not include the units. Please, add the units.

Keywords:

It would be better to add words that have not being cited in the title to improve retrievability, such as “near infrared”, “manure”, et cetera.

Introduction:

Adequate.

Material and methods

Although the multivariate analysis is the best way to analyze NIR data, the authors should present the statistical design of the study and show a univariate analysis of the evaluated variables (table 2).

Results and discussion

3.1. Sample spectra should be substituted with “NIR spectra”.

The NIR spectra figures and discussion is not appropriate. Regarding the figures, it is not clear why there are two vertical axes. It is also not clear the absorbance unit. The authors should present the spectra of each treatment instead of how the spectra were acquired (reflectance or transreflectance). Regarding the discussion, the authors informed the interference of the moisture (OH) content, but they failed to address how they overturned this to develop their models.

The authors should include the units when state the RMSE.

Conclusion:

The conclusion is basically to abstract. The conclusion should state if the hypothesis of the study was accepted or not.

Reference: Check the format of all references.

Figures: The figures are not adequate as they need to add the R2 and RMSEC.

Reviewer 2 Report

The authors satisfactory replied to my observations, the added corrections dispel any doubts about the  spectra representation

This paper can be accepted for publication.

Author Response

Thanks for your review and comment.

This manuscript is a resubmission of an earlier submission. The following is a list of the peer review reports and author responses from that submission.

Round 1

Reviewer 1 Report

Journal: Remote Sensing

Article ID: emotesensing-1488505

Version: 1

Article title: Evaluation of Near-Infrared Reflectance and Transflectance Sensing System for Predicting Manure Nutrients Using Spiking Methodology

Editor: Brigitta Katler

General comments:

            This manuscript describes the use of near infrared (NIR) spectroscopy to develop predicting models for manure attributes, such as ammoniacal nitrogen (NH4-N), organic nitrogen (Org-N), total solid (TS), ash, and particle size (PS). The topic is not novel, but it is interesting and in need with the newest environmental regulations. There are also some aspects that have to be improved in terms of clarity and to make the manuscript appropriate for publication.

I would like the authors to consider the following points.

Specific comments:

  • Abstract:

The authors mentioned “nitrogen species”, but I would recommend “nitrogen speciation”.

The authors used “dairy manure”, I would recommend “dairy cow manure”. Please, substitute throughout the manuscript.

The authors used R2 and RPD to evaluate the models, however, to use of the errors (RMSEC and RMSEP) is recommended in NIR studies. There is also some critics about the using of RPD, but some authors suggest its use.

  • Keywords:

It would be better to add words that have not been being cited in the title to improve retrievability, such as “near infrared”, “manure”, et cetera.

  • Introduction:

The introduction brings general and well-known aspects of NIR spectroscopy and manure analysis. However, the hypothesis of the study and the novelty to the field should be modified as the justification of using spiking to improve calibration models is not strong enough. It would be better to collect more samples than few and add nutrients on it.

  • Material and methods:

The authors should better describe how the samples were separated into calibration and validation set. Did the authors use any algorithm? If only full cross validation was applied, please add this information in the manuscript.

Although excellent authors recommend RPD, many also have criticized its use. In NIR spectroscopy the lower the errors the better, thus the performance of the models is better stated via the errors (RMSEC and RMSEP).

The authors did not present the statistical design of the study. Based on the information, the experiment was set according to a complete randomized design in a factorial arrangement 20 (manure samples) x 4 (1.25, 1.5, 2, and 4 times of concentration in the control) with how many replicates?

  • Results and discussion

It would be great if the authors start this section presenting the NIR spectra. It is also necessary to discuss about the peaks and the spectral features, e.g., the moisture (OH) content related bounds and the N related bounds.

After that the authors would present the concentration results and finally the performance of the models.

I understand that the intent of using NIR spectroscopy was to provide a simple tool for farmers to determine nutrient in fresh manure. However, the high moisture content of manure makes it difficult and the NIR spectra is dominated by the water bounds. The authors should discuss this issue.

Similarly, the authors should present where the N bounds and overtones are located in the NIR spectra and maybe the use of selected wavelengths would help to improve the performance.

Is seems that the authors did not use an external validation/prediction set. It would be great to have the calibration models testes using different batches to show if they are robust.

  • Conclusion:

The conclusion is basically to abstract. The conclusion should state if the hypothesis of the study was accepted or not.

  • Reference: Check the format of all references.

  • Figures: The figures are not adequate as they need to add the R2 and RMSEC.

Concerning all aspects pointed out, I feel that the manuscript is not appropriate for publication as presented, but can be considered after major revision.

Reviewer 2 Report

In this manuscript were evaluated calibration models of ammoniacal nitrogen, total solid, organic nitrogen, ash and particle size, contained in manure samples.

Given the cost of both time and money for chemical analyzes on the samples, it was decided to add ammonium chloride and arginine to widen the variability of the samples in ammoniacal and organic nitrogen, respectively.

In order for the article to be suitable for publication, some parts need to be revised.

Introduction

Pag 2 - The authors state that the purpose of a spiking strategy is to introduce compositional variability into a basic raw calibration dataset and expand prediction models without collecting more samples, and report that this strategy was used for soil analysis. Meanwhile, as for soils, the spiking strategy is to add "local" samples to improve "global" calibration. In practice, a personalized calibration is built, it is certainly not to expand the variability of the composition using fewer samples.

Materials and methods

2.2 Manure Spiking – Have the authors analyzed some spiked sample by chemical analyses to verify the affective ammoniacal and organic nitrogen content?

2.4 Statistical analysis and PLS calibration assessment – How many samples were considered in the venetian cross validation for each segment?

Results and discussion

Figure 1 and 2 – The depicted PB spectra shown in the figures are definitely not expressed as absorbance but shown as reflectance (i.e. transflectance). Therefore, it is wrong to state that reverse trends of the absorbance were observed in transflectance mode. I suggest the authors check how the spectra were originally recorded if in reflectance or directly in absorbance mode.

Is something missing in the sentence: “The NH4-N and K were highly correlated to TN content and were N-group parameters which was comparable to the results in this study.”? Because the meaning is not understood.

 3.2.1. Path lengths

The PB1 sensor has better predictive capabilities because its spectrum is less saturated, and not because it has a better signal-to-noise ratio.

3.3. Relationship between nitrogen and other manure parameters

How can the authors explain such a low accuracy in predicting N in some SG samples (R2=0.014 and 0.00 for the NH4-N spiked group, and R2=0.149 for the Org-N spiked group) ?.

Round 2

Reviewer 2 Report

After the suggested correction the manuscript was improved.

I'm sorry to bother the authors but I would like to point out that Figures 1 and 2 are incorrect. The problem is not the y-axis, but the shape of the transflectance spectra, as they are not shown as absorbance mode (log 1/R or log 1/T) but as reflectance mode (R o T). I suggest the authors to see reference 16  figure 3  where the tranflectance spectra are correctly depicted.

If the authors had considered, for subsequent elaborations, the reflectance spectra as if they were absorbance spectra, they would have to recalculate them.

There is a typo, both in the abstract and in the text (pag 4): RMSE = root mean square error not room mean square error.